# Virtual Reality Digital Twin and Environment for Troubleshooting Lunar-based Infrastructure Assembly Failures

Phaedra S. Curlin
*Center for Astrophysics and Space Astronomy*
*University of Colorado Boulder*
Boulder, CO, USA
phaedra.curlin@colorado.edu

Madaline A. Muniz
*Center for Astrophysics and Space Astronomy*
*University of Colorado Boulder*
Boulder, CO, USA
madaline.muniz@colorado.edu

Mason M. Bell
*Center for Astrophysics and Space Astronomy*
*University of Colorado Boulder*
Boulder, CO, USA
mason.bell@colorado.edu

Alexis A. Muniz
*Center for Astrophysics and Space Astronomy*
*University of Colorado Boulder*
Boulder, CO, USA
alexis.muniz@colorado.edu

Jack O. Burns
*Center for Astrophysics and Space Astronomy*
*University of Colorado Boulder*
Boulder, CO, USA
jack.burns@colorado.edu

*Abstract*—Humans and robots will need to collaborate in order to create a sustainable human lunar presence by the end of the 2020s. This includes cases in which a human will be required to teleoperate an autonomous rover that has encountered an instrument assembly failure. To aid teleoperators in the troubleshooting process, we propose a virtual reality digital twin placed in a simulated environment. Here, the operator can virtually interact with a digital version of the rover and mechanical arm that uses the same controls and kinematic model. The user can also adopt the egocentric (a first person view through using stereoscopic passthrough) and exocentric (a third person view where the operator can virtually walk around the environment and rover as if they were on site) view. We also discuss our metrics for evaluating the differences between our digital and physical robot, as well as the experimental concept based on real and applicable missions, and future work that would compare our platform to traditional troubleshooting methods.

*Index Terms*—telerobotics, lunar robots, human-robot collaboration, digital twin, virtual reality

## I. Introduction

NASA is working to create a sustainable human presence on the lunar surface by the end of the 2020s. This includes the construction of the Lunar Gateway, a habitat and science laboratory, that will orbit the Moon starting in the middle of the decade. In addition to direct operation from Earth, the Gateway will enable astronauts to use low-latency communications with the lunar surface (especially the far side) assets as well as teleoperate rovers across the Moon to perform tasks. To aid in creating this sustainable presence on the Moon, humans and robots will need to collaborate to carry out complex tasks like assembling *in situ* resource utilization or radio telescopes, such as for Farside Array for Radio Science Investigation of the Dark Ages and Exoplanets (FARSIDE) [1]. This mission will require the rovers to autonomously deploy a low frequency interferometric array on the far side of the Moon. In the case of an autonomous failure, such as a misaligned antenna, lunar telepresence would allow astronauts to recover the rovers from the state of failure.

Current troubleshooting methods for rover recovery can be seen with the Mars Yard rover full-scale, engineering model twin. The Yard also emulates Martian terrain in terms of soil characteristics and obstacles (e.g., boulders) [2]. While this Earth-based physical twin and environment can be modified to help simulate encountered issues, drawbacks include little portability, difficulty in simulating gravity, and no method of creating many replicas of the equipment and rovers themselves.

Novel technologies could also be used to aid with the troubleshooting process to recover these rovers. This includes stereoscopic cameras which allow for stereoscopic passthrough and 3D point cloud generation. Such cameras could be mounted on the rover, giving astronauts the ability to see depth from the perspective of the rover or interact with a 3D model of the environment. Virtual reality (VR) headsets could be used in conjunction with the 3D models to allow for the astronauts to walk around in the reconstructed environment (Fig. 1).

We have devised a VR platform that will allow operators to troubleshoot the rovers in a risk-free environment. The operators will be able to interact with a digital twin of the rover that uses the same kinematic model, control interface, and hardware model, as well as walk around in a high-fidelity reconstruction of the local environment. The operators will then apply the solutions developed in the VR space by teleoperating the rover.

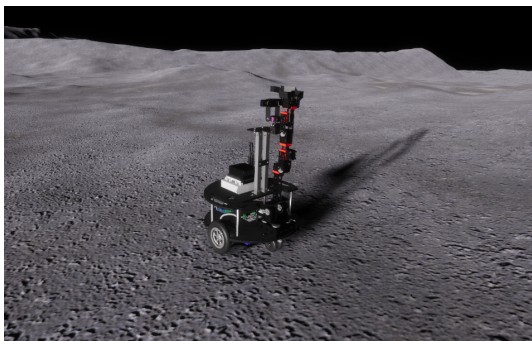

Fig. 1. The "Armstrong" rover on the simulated lunar surface in Unity. The digital twin can be placed in any environment of choice. Teleoperators can walk around in the environment and interact with the rover.

In this paper, we first describe the physical design of our robot. We then discuss the design process for the digital twin and its environment. The third portion of our paper will show our plan to evaluate the fidelity of the digital twin compared to the physical rover. The remainder of the paper will present our experiment and its methodology, as well as our near- and long-term plans regarding the experiment.

## II. PHYSICAL ROVER

Our physical rover, nicknamed "Armstrong", is a Parallax Arlo Robot System with two motorized wheels. A CrustCrawler Pro-Series six degrees of freedom (DoF) mechanical arm has been mounted on the rover. This allows for the teleoperator to drive the rover and grasp objects. "Armstrong" also has a ZED Mini stereoscopic camera on servo motors at the top of its mast. Thus, the user has the ability to see through the "eyes" of the rover as well as manipulate the camera rotation to mimic head movements. "Armstrong" is equipped with a Jetson Xavier AGX running the Robot Operating System (ROS) [3] to control it with an Xbox 360 gamepad (Fig. 2, Fig. 3).

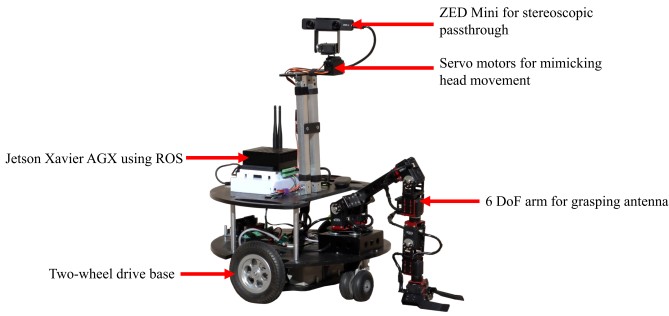

Fig. 2. The "Armstrong" rover. Its robotic arm allows for it to manipulate items such as antenna nodes in its environment. It is also equipped with a ZED Mini to give the operator a first-person perspective from the rover.

## III. VIRTUAL REALITY DIGITAL TWIN AND SIMULATED ENVIRONMENT

The virtual reality platform was designed in Unity [4], a video game engine that also simulates physics. This allows us

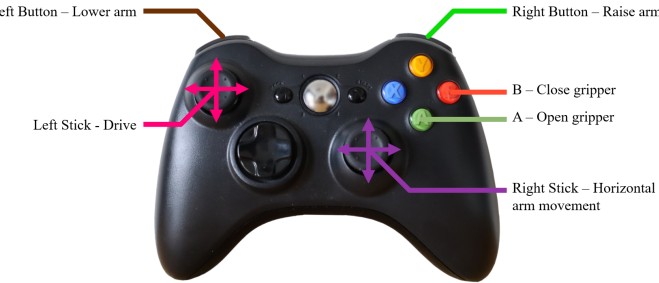

Fig. 3. "Armstrong's" Xbox 360 button mapping. This popular gamepad makes "Armstrong's" control feel more intuitive to the operator.

to import 3D models and use VR headsets, like the Oculus Quest. In the following two subsections we discuss the design process behind generating the digital twin and simulated environment. We then explore the different features that the VR platform includes alongside the twin and environment.

### A. Digital Rover Design

Our digital twin of "Armstrong" uses the same computer-aided design (CAD) model as the ones used for the physical rover's design. This generates the geometry required for the collision model of the rover which dictates how it interacts with surfaces in the environment. Currently, there are no simple methods to export the models to the correct file format required for Unity. Furthermore, CAD models also do not support high resolution material texturing which allows for more realistic looking designs. For these reasons, we exported the CAD model to a free and open-source 3D-modeling software, called Blender [5]. Here, we applied photorealistic textures and exported "Armstrong's" model (Fig. 4) to the required format by Unity.

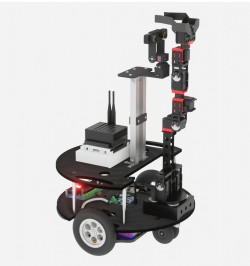

Fig. 4. Blender render of the "Armstrong" rover. This model is based on the CAD design for the rover and conserves the collision model. Each component was textured to increase the visual fidelity of the model.

The 3D model is imported using the Unity Robotics Hub [6] suite which also gives the ability to control the model using the ROS interface along with a full collision model. In our case, we used the same controls as the ones used by the physical rover.

### B. Digital Environment Design

The virtual environment is a reconstruction of the space in which the participants will carry out the experiment. To do so, scans of the environment were taken using the ZED Mini.

However, this yielded a highly distorted render of the space. As a workaround to this issue, we gathered measurements of the experiment room as it has a simple layout. We photographed the room to generate photorealistic textures of the walls and items located in the room. Similarly, to the digital twin design, we created the model of the room in Blender. This reconstruction of the room will serve as the equivalent to a high-resolution scan of the environment (Fig. 5).

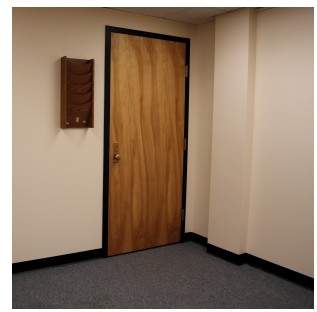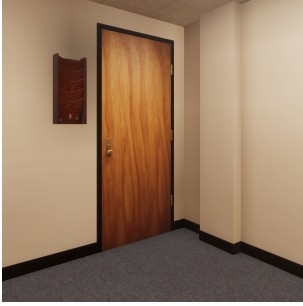

Fig. 5. The physical experiment room (left) and the virtually recreated experiment room in Unity (right). This photorealistic recreation of the room was created using measurements and photographs. This gives the operator an increased impression of being present at the site of the rover.

Material properties can also be set in the environment such as static and dynamic friction. This is especially important for the carpeted floor of the experiment room as it always has direct contact with the rover. The static friction of the floor will be calculated by using a force gauge with an object of a known mass that will be dragged across the floor. The dynamic friction will be calculated by driving the rover from a state of rest to a known velocity over a predetermined time interval. Since the rover is made of materials that are standard, like aluminum and rubber, we will use common values found for static and dynamic friction.

Gravity can be set to whatever value the user desires. Since the experiment room is located on Earth, the environment's gravity, $g$, will be simply set to $g = -9.81ms^{-2}$.

However, it is important to note that the physics properties (i.e., friction coefficient and gravity) provided by Unity are uniform. Custom scripts will be added to increase the physical properties of the digital environment.

### C. Virtual Reality Platform

As mentioned previously, the VR platform uses the Unity video game engine. This allows us to create custom VR environments compatible with a variety of VR headsets, including the Oculus Quest. This headset tracks the user's position and head movements. These statistics are passed onto Unity through a "rig" that recreates the actions performed by the user and that gives them the ability to see into the virtual world (Fig. 6).

This rig can be used for an exocentric point of view where the user can walk around in the environment and interact with and control the rover. We have implemented an egocentric point of view that mimics the stereoscopic passthrough (Fig. 7). Virtual cameras located in the model of

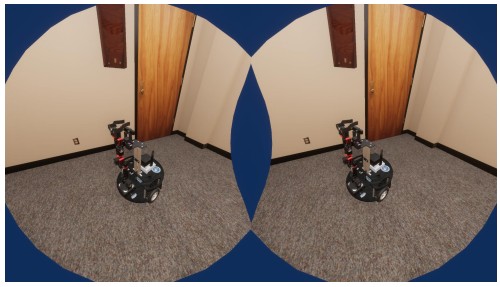

Fig. 6. Ocular view "Armstrong" from the point of view of the teleoperator. Each eye has a separate video stream creating the illusion of depth with the VR headset. This gives operators the impression of being *in situ*.

the ZED Mini mimic the stereoscopic passthrough and have the similar properties such as for the field of view (i.e., 110°) and maximum movement angle set by the servos (i.e., 180° horizontally and vertically).

An additional feature that we have implemented for the VR platform is the ability to reset the environment at the press of a button on the gamepad. The user will be able to revert to the initial state of the world (as it was when the operator was first introduced). The ability to restart is one of the main components that makes the platform risk-free.

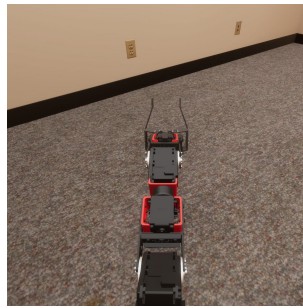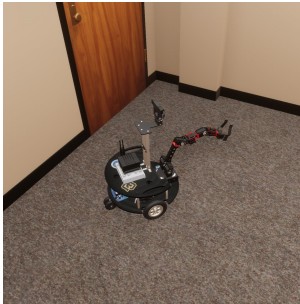

Fig. 7. The egocentric perspective (left) and the exocentric perspective (right) in Unity. The prior imitates the way in which the operators can see through the stereoscopic camera. This gives them a view from the robot's perspective in the environment, in terms of height and position. The latter imitates the way in which a teleoperator can walk around a physical twin with the added benefit of being in a reconstruction of the environment.

### D. Evaluating Virtual Model Accuracy

Since we are attempting to recreate the physical world in a virtual one, we must evaluate how accurate our virtual model is. It is important to note that environmental inaccuracies (e.g., due to using uniform gravity, dynamic friction, and static friction) are harder to address as they require additional scripting in the game engine. Limitations regarding the digital twin include changes in the collision model of the rover (e.g., a component falling off, parts becoming severely damaged) and component wear (e.g., torque power of motors reducing over time). The digital twin can however be calibrated to match the physical model more closely. This will be done by looking specifically at four attributes:

- *ROS message latency:* This will measure the time delay for receiving joint position and driving command mes-

sages from ROS to the rest of the rover. This will be compared to the latency between ROS and the digital twin in Unity. If the physical rover message latency is slower than the digital twin, messages can be delayed to match the real latency.

- *Absolute joint error:* This will evaluate the difference in angle that we expect to measure as opposed to the one read by the arm. If the physical arm has a greater error value than the digital arm, parameters like arm stiffness can be reduced to generate similar absolute joint error values.
- *Joint movement resolution:* Documentation for the arm provides the smallest possible step that can be executed (i.e., 0.0888°). A small joint movement step can be executed in the digital world. The digital rover's joint resolution can be modified by changing the minimum tolerable step that is permitted by ROS to match the physical arm's.
- *Driving motion errors:* Since the virtual and physical rovers can simultaneously receive the same input driving commands, we can evaluate if the two models will traverse the same distance. This will be influenced by, for example, the friction at the interface between the wheels and surface. This can be mimicked by the digital model by first changing the friction parameters of the floor. Torque values being applied to the wheels can be changed accordingly depending on the slip that is measured on the physical rover.

For each of these measured metrics, adjustments can be made in the digital twin to match up the properties of those in the physical rover.

## IV. PROPOSED EXPERIMENT

### A. Experimental Concept

Our experiment was designed to be applicable to real missions such as FARSIDE. Since in the future, we would like to compare the use of a digital twin for troubleshooting as opposed to a physical twin, we will first carry out a baseline experiment to evaluate our VR platform. This means comparing participants using the VR platform for troubleshooting to none at all and evaluating their performances.

For this experiment, the participants will be tasked with reorienting misaligned dipole antennae with the rover (Fig. 8). These antennae will have a known target orientation. The teleoperators will have the ability to troubleshoot and experiment with ideas in the virtual environment and then apply their knowledge to the physical rover.

### B. Methodology

The experimental methodology is in the early stages of development. The current plan for the experiment that we plan to conduct later this year will consist of two groups of participants: Group A which will develop ideas in the VR digital twin and environment then teleoperate the rover into recovery, and Group B which will not use the VR digital twin

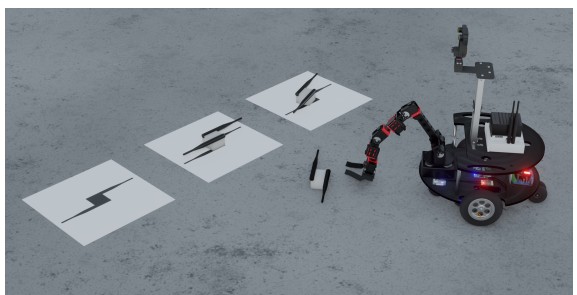

Fig. 8. Blender render of the "Armstrong" rover experimental concept. The antenna outline on the mats will indicate the required orientation for participants to carry out the antenna realignment task.

and environment and will be required to directly teleoperate the rover for its realignment task (Fig. 9).

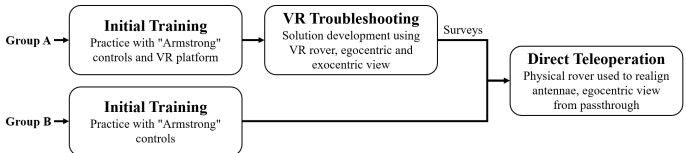

Fig. 9. Experimental methodology. This will be a baseline experiment to evaluate the effectiveness of use the VR platform as opposed to none at all for troubleshooting the antenna realignment task.

The Group A participants will be introduced to the experiment with initial training. They will have an overview and practice using the controls, as well as be familiarized with the VR platform. Next, they will go into the virtual recovery platform and begin the task there. They will have a unique view of the task with ego- and exo-centric views with the ability to control "Armstrong". Following their completion of the task in the VR platform, they will receive the surveys. They will then go into direct teleoperation and complete the same task, only this time with the physical rover to apply their developed solutions.

For the Group B participants, they will also receive an initial training which will consist of an overview and practice time for controlling the rover. However, unlike group A, they will directly attempt to realign the antennae without having the ability to develop and troubleshoot in the VR environment.

### C. Measures

We have established variables that we will evaluate and determine the effectiveness of our VR digital twin and environment for troubleshooting based on methods for evaluating robot accuracy [7], [8]. The subjective metrics will consist of between participant surveys to see how their decision-making process is affected:

- *Situational awareness*: will evaluate how well the participant understands their environment, which is needed for missions in which teleoperators cannot be *in situ*. This can be analyzed using methods like the Situation Awareness Global Assessment Technique (SAGAT) [9]

or Location, Activities, Surroundings, Status, and Overall (LASSO) [10] technique.

- *Cognitive load*: will show how the difficulty of the task will affect their process. This can be evaluated using surveys like the NASA Task Load Index (NASA TLX) [11] or the Bedford Workload Scale [12].
- *System usability*: will indicate how usable our VR digital twin and environment are. This can be evaluated using the System Usability Scale (SUS) [13].

The performance metrics will give us statistical data to measure the validity of the VR environment:

- *Time to completion*: will measure the time taken by the participant to realign the antenna and will evaluate how successful the participant was in executing the task. This will be used in the virtual and physical environment.
- *Number of resets*: will measure how many times the VR world was reset back to the initial state. This will evaluate the usefulness of the "reset world" feature in the VR environment.
- *Success rate over X attempts*: will indicate how many antennae were realigned correctly. This will be a second indicator to showing how successful the participant was in executing the task. This will be used in the virtual and physical environment.

## V. CONCLUSION AND FUTURE WORK

In this paper, we proposed using a VR digital twin and environment to aid teleoperators in troubleshooting rovers on the Moon's surface, similarly to how physical twins are currently used. This VR digital twin uses the same controls and kinematics as does the physical rover. The operator can be in the egocentric and exocentric perspectives of the rover by leveraging novel technologies such as stereoscopic cameras. A VR reconstruction of the environment can also be implemented to simulate the surroundings. This gives the teleoperator the ability to virtually walk around and interact with the rover.

Our short-term plans include measuring how equivalent our digital twin is to the physical rover by looking at ROS message latencies, absolute joint error, joint movement resolution, and driving motions error. Since we expect the physical twin to be less accurate, we will calibrate our digital twin to the physical rover's parameters. We will also develop surveys for measuring teleoperator's situational awareness, cognitive load, and the system usability of our VR platform based upon best practices. Finally, we will run a preliminary test experiment to see if there are flaws that have been overlooked in our platform and methodology.

In the long-term, we plan to compare differences in using a physical twin as opposed to a digital twin. This would be an in-depth comparison between our newly proposed model and the traditional methods that are currently used. Additionally, more complex tasks could be tested such as tether deployments which would require introducing rope physics.

## ACKNOWLEDGMENT

This work was directly supported by the NASA Solar System Exploration Virtual Institute cooperative agreement 80ARC017M0006.

We would like to thank members of the NASA Solar System Exploration Research Virtual Institute's Network for Exploration and Space Sciences NESS team including Daniel Szafir, Michael Walker, Terry Fong, and Lockheed Martin Corporation for their valuable help and guidance.

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
