# OpenReview forum: "Virtual Reality Digital Twin and Environment for Troubleshooting Lunar-based Infrastructure Assembly Failures"
_humanrobotinteraction.org/HRI/2022/Workshop/VAM-HRI — VAM-HRI 2022_

### Official Review · Reviewer_BzGK · 2022-02-23
**Good System Paper with Initial Study Design**

**Rating:** 8
**Confidence:** 5

**Review:**

Summary:
This paper proposes a virtual reality (VR) "twin" for the "Armstrong" rover as a means to diagnosis problems/help teleoperate the Rover during space missions. The design of the twin + system are discussed with future work introduced (e.g., designing more complex physics for the twin system). A between-subjects study is presented to evaluate the efficiency/efficacy of the system. Overall the paper is well written and would be a great contribution for VAM-HRI.

Feedback:
The system seems cool and very interested to see the physics tuning with the real robot. I think the paper is generally well written. Understanding/simulating more complex physics is not something I have seen a lot of work on in VAM-HRI so I hope other people are as interested as I am.

The largest room for improvement is around study design (see * comments below for details). I also believe the system could benefit from background design considerations from other VAM-HRI work for teleop/signalling (I'm sure many people at the workshop can provide some great resources/discussions around this). The feedback marked below has a lot of nitpicks so apologies if they come off rude in any way, only suggestions for improving the system/paper.
 - if done in LaTex, change your quotation marks from " to `` for opening and '' for closing (will orient them correctly)
 - you refer to an (or a? not sure which is correct) Xbox 360 controller and the figure which would make me assume there is a controller in the picture. I would either move the Fig reference up or add an Xbox controller in the picture.
 - did you use ROS# or Unity Robotics Hub to import the URDF? If ROS#, I would add a quick reference
 - if you are referencing Unity, I would reference Blender or say it is a free (or open-source? I forget and currently don't have internet) 3D modeling software
 - do you have an idea of how many participants you are going to run for each group/plans for recruitment? I think the latter matters a lot given whatever the real world application is (experienced/inexperienced with robots/vr/games)
 - for describing the study set up, a diagram may be helpful
 - *for A vs B, is it:
    - A: Training -> VR task -> surveys -> real world -> surveys
    - B: Training (without VR) -> real world -> surveys
 - *if so, there is a potential for bias in total time spent with the robot. Group A can potentially* just have more time to understand the robot. To potentially avoid this, I would either basically make this a within subjects study (make Group B do Training -> physical world -> surveys -> VR Training -> VR Task -> surveys [not perfect]) or [I think a better approach] make group B have a more traditional interface/similar activity to VR (basically whatever is used now). I guess it depends what you want to measure/isolate, just more difficult to make the claim "Our VR interface is better than no interface" when the interface team just has more time with the activity
 - possibly the above is me misunderstanding the physical vs virtual twin

---

### Official Review · Reviewer_26GJ · 2022-02-25
**Interesting and novel work, clear accept**

**Rating:** 8
**Confidence:** 5

**Review:**

The proposed virtual reality digital twin for lunar-based teleoperation is novel and interesting and relevant to the VAM-HRI community. This paper is also well-written and future work is clear, so this is a clear accept. The reviewer's suggestions are mostly focused on writing, and the reviewer looks forward to the implementation of the future work.

1. In abstract, the word “view” should be in the fifth sentence.
2. Equation 1 describing the gravity constant probably can be inline rather than be a separate equation.

---

### Decision · Program_Chairs · 2022-03-04

Accept